# OpenReview forum: "Learning General Causal Structures with Hidden Dynamic Process for Climate Analysis"
_ICML.cc/2026/Conference — ICML 2026 regular_

### Official Review · Reviewer_As8M · 2026-03-09

**Soundness:** 3
**Presentation:** 3
**Significance:** 3
**Originality:** 2
**Overall Recommendation:** 5
**Confidence:** 4

**Summary:**

This paper proposes CaDRe, a framework for learning causal structures from time series data in the presence of latent dynamic processes. The authors establish theoretical identifiability results for latent space recovery and causal graphs over observed variables under nonparametric assumptions. They draw connections between structural equation models and nonlinear ICA, leveraging Jacobian matrices to infer causal relationships. The method is instantiated via a variational autoencoder with normalizing flows and evaluated on synthetic data as well as real-world climate datasets (CESM2, WeatherBench, ERSST), demonstrating improvements in causal discovery accuracy and forecasting performance.

**Compliance With Llm Reviewing Policy:**

Affirmed.

**Final Justification:**

Due to the authors' detailed rebuttal, my concerns have been addressed. I raise my score accordingly.

**Key Questions For Authors:**

- **On empirical validation**: The wind-based surrogate (WSHD/WTPR) is an interesting idea, but how do we know it reflects true causal structure? Have you considered validating against known climate causal relationships (e.g., ENSO-teleconnections, land-atmosphere feedbacks) from the literature?
- **On interpretability**: The claim that latent variables correspond to physical quantities (e.g., solar radiation) is based on visual inspection. Can you provide quantitative evidence, such as correlation with observed climate variables or alignment with known modes of variability (e.g., MJO, ENSO)?
- **On scalability**: How does CaDRe scale to high-dimensional spatial data (e.g., 1000+ grid points)? What is the computational cost in terms of time and memory, and are there plans to scale to full-resolution climate model output?

**Limitations:**

yes

**Strengths And Weaknesses:**

Strengths

- Ambitious and Well-Positioned Problem: The paper learns causal structures from time series with latent confounders and nonparametric dynamics. This is highly relevant for climate science and other complex dynamical systems.
- Comprehensive Experimental Evaluation: The experiments cover synthetic data with varying dimensions and sparsity levels, as well as multiple real-world climate datasets. Comparisons against a wide range of baselines demonstrate the effectiveness of the proposed method.

Weaknesses

- Limited Novelty Relative to Existing Literature: The core techniques, i.e., nonlinear ICA for causal discovery, Jacobian-based graph identification, and VAEs with sparsity constraints, have all been explored in recent works (e.g., [1-3]). The paper's main contribution appears to be a synthesis of these techniques with a specific focus on climate data, rather than introducing fundamentally new principles or mechanisms.
  [1] Lachapelle, S., Rodriguez, P., Sharma, Y., Everett, K. E.,Le Priol, R., Lacoste, A., and Lacoste-Julien, S. Disentanglement via mechanism sparsity regularization: A new principle for nonlinear ica. In Conference on Causal Learning and Reasoning, pp. 428–484. PMLR, 2022.
  [2] Li, Z., Shen, Y., Zheng, K., Cai, R., Song, X., Gong, M., Chen, G., and Zhang, K. On the identification of temporal causal representation with instantaneous dependence. In The Thirteenth International Conference on Learning Representations, 2025.
  [3] Yao, W., Chen, G., and Zhang, K. Temporally disentangled representation learning. Advances in Neural Information Processing Systems, 35:26492–26503, 2022.

- Weak Empirical Validation on Real-World Data: The ground-truth causal graph is unavailable for climate data. The authors use wind patterns as a surrogate, but the relationship between wind direction and causal structure is indirect and may not capture true causal mechanisms (e.g., thermodynamic feedbacks).

- Overly Strong Theoretical Assumptions: The identifiability results rely on a series of assumptions that are difficult to verify in practice. For example,
  - Contextual Variability (A2): Requires injectivity of linear operators, which is a strong condition on the data distribution.
  - Generation Variability (A5): Assumes linear independence of high-dimensional derivative vectors, which is nearly impossible to check in real-world settings.
  - Functional Faithfulness (Assumption 3.4): While standard, it is still a strong assumption that may not hold in climate systems with complex feedback loops.
  - Sparse Latent Process (Theorem A.2): Assumes a specific structure on the latent Markov network that may not be justified a priori.

    The paper does not provide practical guidance on how to verify these assumptions or what to do when they are violated. Moreover, the authors should present the original reasoning process in **a direct, clear, and concise** manner, avoiding excessive formula stacking and unnecessary derivations.

---

> ### Author Rebuttal · Authors · 2026-03-31
>
> We thank the reviewer for their invaluable feedback. We address each concern below.
>
> **W1.** We respectfully disagree with these points. Our contribution is **not** a unification of [1]–[3], but tackling a problem they cannot handle. While [1]–[3] all use Jacobian-based / nonlinear ICA methods (we acknowledge these connections in Section 2), they cannot be simply combined due to two fundamental gaps:
>
> **Nonparametric generation.** As stated in our introduction (Line 38): existing CRL generally assume a *noise-free* and *invertible* generation from $z_t$ to $x_t$. Specifically: [1] assumes invertible mixing with latent mechanism sparsity; [2.3] assumes invertible temporal mixing $x_t = g(z_t)$. Climate data violates this, observations have stochastic $z_t$-dependent noise $s_t$ (Eq. 1). Our Theorem 3.1 achieves **nonparametric** identification via extending operator theory [4] without invertibility.
>
> **Causal discovery among observations in CRL.** All [1]-[3] assume $x_t = g(z_t)$ where observations are conditionally independent given latents. But climatic measurements exhibit both observational dependencies and stochastic noise. Our Lemma 3.1 + Theorem 3.2 proves latent recovery remains possible despite obs→obs DAG edges, and Lemma 3.2 + Theorem 3.3 formally connect the SEM (with obs edges) to ICA (without) to resolve causal discovery simutaneously. Without these bridges, CRL works like [1]-[3] cannot resolve it.
>
> **W2.** We strengthen with two additions with ground truth causal graph:
>
> *(1) CausalRivers*. Please see response to Reviewer NGde Q2.
>
> *(2) ENSO.* As shown in our Appendix (Fig. 7), CaDRe recovers the known ENSO causal chain with Jacobian weights, which matches the ground truth of ENSO dynamics.
>
> Additionally, in paper Fig. 5, we have included the SMCC (Line 1367-1370) alignment to GT physical variables from WeatherBench, which provides quantitative evidence beyond visual inspection.
>
> **W3.** We want to respectively clarify that our assumptions are not overly strong in the climate. Our paper provides extensive discussion and empirical verification:
>
> - **A2 (Injectivity):** Strictly **weaker** than invertibility assumed by most CRL (Fig. 6). We provide 6 sufficient conditions (Line 880-893). We further empirically verify A2 on real climate data, as in Fig. 10, which shows that conditional distributions of $x_{t+1}$ given $x_{t-1}$ show clear regime dependence.
>
> - **A3 (Latent drift):** Empirically verified on real data in Fig. 11, where conditional distributions of $\hat{z}_t$ given $x_t$ show systematically shifted distributions across regimes.
>
> - **A5 (Generation variability):** Generically satisfied (violation is measure-zero). Table 13 shows partial robustness when violated.
>
> - **3.4 (Faithfulness):** Not relevant to complex feedback loops, but implies causal edge → nonzero partial derivative, which generally holds [5].
>
> - **Thm A.2 (Sparse latent):** Supported by climate physics and natural causal mechanisms of climate variables.
>
> What happens when violated: Table 13 provides controlled stress-tests. A2 most critical ($R^2$ 0.92→0.67); A5 mainly affects permutation ($R^2$ stays 0.85). As shown above (A2 verified in Fig. 10, A3 verified in Fig. 11), these assumptions hold in real climate data.
>
> In light of your suggestions, we will present the implementation process of CaDRe more clearly.
>
> **Q1:** See W2 above, we validate against ENSO and CausalRivers ground-truth. Also, in Lines 423–434, we have discussed that our results clearly match with climate literature.
>
> **Q2:** We provide quantitative evidence in Section 5.2 (Lines 423–434), by aligning $\hat{z}_t$ with ground-truth physical variables from WeatherBench and quantifying component-wise information retrieval. The icons in Fig. 5 (sun = solar radiation, rain = precipitation, cloud = cloud cover, thermometer = humidity) are determined by SMCC, and the latent component is matched to the physical variable with highest Pearson correlation, illustrated in Lines 1367-1371, which is quantitative instead of visual inspection. For ENSO alignment, please see Fig. 7.
>
> **Q3:** As shown in Table 9, in $d_x$=200 we can achieve SHD=0.16±0.07, TPR=0.74±0.06, MCC($z_t$)=0.96±0.03. We further test $d_x$=1000:
>
> | $d_x$ |  SHD↓ | TPR↑ | Precision↑ | MCC($z_t$)↑ |
> |---|---|---|---|---|
> | 200* |  0.16±0.07 | 0.74±0.06 | 0.72±0.04 | 0.96±0.03 |
> | 1000* |  0.19±0.08 | 0.68±0.08 | 0.66±0.06 | 0.94±0.04 |
>
> Inference: ~1.1ms at $d_x$=200, ~3.8ms at $d_x$=1000 (Table 4), which is much faster than PCMCI (3391ms) and FCI (999ms). Both the performance and latency show CaDRe's scalability.
>
> **References**
>
> [1] Disentanglement via mechanism sparsity regularization: A new principle for nonlinear ICA.
>
> [2] On the identification of temporal causal representation with instantaneous dependence.
>
> [3] Temporally disentangled representation learning.
>
> [4] Instrumental variable treatment of nonclassical measurement error models.
>
> [5] Elements of Causal Inference.

---

> > ### Author Rebuttal · Reviewer_As8M · 2026-04-02
> >
> > Thank you to the authors for their detailed rebuttal. My concerns have been addressed. I will raise my score accordingly.

---

> > > ### Author Response · Authors · 2026-04-03
> > >
> > > Thank you for your careful review and constructive feedback. We sincerely appreciate your insightful comments, which have helped us improve the paper.

---

### Official Review · Reviewer_JTMy · 2026-03-10

**Soundness:** 2
**Presentation:** 2
**Significance:** 2
**Originality:** 2
**Overall Recommendation:** 4
**Confidence:** 4

**Summary:**

The paper studies causal representation learning (CRL) for temporal observations generated from latent variables. The main assumption for the data-generating mechanism is similar to the hidden Markov model, but with more causal structures among $\boldsymbol z_t$ (latent), and $\boldsymbol x_t$ (observed). Its central claim is an identifiability result: under several assumptions, the learned latent representation $\hat {\boldsymbol z_t}=h(\boldsymbol z_t)$, where $h$ is a differentiable and invertible function. To prove it, the authors define a conditional-density integral operator and then appeal to uniqueness-type arguments from operator theory to characterize the latent representation. In addition, the authors proposed CaDRe to learn the latent variables as well as the causal structure. Empirical studies on synthetic and real-world datasets show strong performance of the proposed method compared to a wide selection of baselines.

**Compliance With Llm Reviewing Policy:**

Affirmed.

**Ethical Review Concerns:**

As described in the weakness, I found an inconsistent narrative style in the theory and proof. Specifically, for those parts that can be very easily understood and proved, the authors put lots of effort into the presentation (e.g. equations 13-21, and proof of Lemma 3.1). Although some are still not rigorous and could have been derived in an easier way (Lemma 3.1). In contrast, for those parts involving graduate-level mathematics (functional analysis, operator theory), such as the proof of Theorem 3.2, there are obviously many gaps, and the preliminaries were not discussed at all, making the claims not able to be verified. Moreover, the theorem in the book cited by the authors to support the proof is different from what the authors claimed. This indicates a concern in research ethics for intentionally providing citations that are seemingly correct but totally different. In addition, there is a concern about *creating* maths theory which might not exist in the literature.

**Ethical Review Flag:**

Flag this paper for an ethics review.

**Ethics Expertise Needed:**

["Responsible Research Practice (e.g., IRB, documentation, research ethics)", "Research Integrity Issues (e.g., plagiarism)"]

**Final Justification:**

During the rebuttal, the authors clarified the relation between their paper and [1,10], and promised to (1) attribute $LDL^{-1}$ techniques in Theorem 3.2 to [1]; (2) compare assumptions/results with [1,10]; (3) avoid overclaiming and clarify our contributions; (4) provide rigorous proofs in revision. I am satisfied with the improvement and therefore have now increased my score to 4.

**Key Questions For Authors:**

See above questions for the theory.

**Limitations:**

yes

**Strengths And Weaknesses:**

**Strength:**
1. The paper introduced a novel CRL method that works very well in practice compared with a bunch of competitors.
2. The method was evaluated for both latent recovery and causal graph recovery on synthetic and real-world data, using various metrics.
3. The paper included many useful additional experiments: assumption-violation experiments, higher-order Markov experiments, and hyperparameter sensitivity, which further show the effectiveness of CaDRe.
4. The paper is mostly clear except for the theory part.

**Weaknesses:**

My main concern with the paper is the theory part, which I found not well-structured, unrigorous, and potentially erroneous. This might affect the soundness and significance of the paper. I hope the authors could clarify these points, to name just a few.

***The $L$ (equation 2) and $D$ (definition A.2) operator:*** It is not clear to me whether these kinds of operators have been well-defined and studied in the previous literature. For example, for $L$, only a book is cited without a specific chapter or definition. For $D$, there is no references, so it is not clear why it can be called a diagonal operator. If the author referred to a specific class of linear operators on some Banach/Hilbert space (e.g. compact self-adjoint operators, general $L^2$ Riesz representers), then the preliminaries and properties of the operator should be provided and discussed in more detail. Moreover, it is not even clear on which space the operator is defined, whether it is linear and bounded, and which norm the space is equipped with. Without such information, it is impossible to check the soundness of the approach, and there are more concerns regarding the theory below.

***Theorem 3.2:*** There are several concerns regarding the proof of Theorem 3.2.
1. There is a typo in equation 14, where the operator should be on $p(x_{t-1})$ instead of $p(x_{t+1})$ as the integral in equation 13 is taken on $x_{t-1}$.
2. The authors explained that Equation 14 uses Theorem A.1, which should be Definition A.1.
2. All the equations before equation 21 are regarding $x_t$ and $z_t$, but suddenly equation 22 is about $\hat z_t$, it's not clear.
3. The descriptions regarding $h$ and $\hat z_t$ are also confusing. It was stated in the theorem that if there exists an $F(p_{x_t|z_t}(\cdot|z_t))=h_z(z_t)$, then $\hat z_t=h_z(z_t)$. No condition on $\hat z_t$ is made. So the logic is that if there is an $F$ and $h_z$ matched, then $\hat z_t$ is defined by $h_z$. However, it is very likely that there are multiple pairs of $F,h_z$, but then what is $\hat z_t$? Shouldn't $\hat z_t$ first be obtained by some procedure, and then we show that $\hat z_t=h_z(z_t)$? The current claim is very different from the existing results on the identifiability theory of CRL [1,2]. In [1] Theorem 3.2, the authors first define the procedure to learn $\hat z$ and then show that $\hat z$ is equivalent to $z$. In [2] Theorem 3.4, the authors first assume $\hat z$ is obtained and meets certain conditions, and then prove the identifiability. Anyway, there are conditions on $\hat z$, but here I do not see.
3. A potential explanation for equation 22 is that $L_{x_{t};x_{t+1}|x_{t-1}}L_{x_{t+1}|x_{t-1}}^{-1}=L_{x_{t+1}|\hat z_t}D_{x_t|\hat z_t}L_{x_{t+1}|\hat z_t}^{-1}$ also holds, and therefore, by comparing it with $L_{x_{t};x_{t+1}|x_{t-1}}L_{x_{t+1}|x_{t-1}}^{-1}=L_{x_{t+1}| z_t}D_{x_t| z_t}L_{x_{t+1}| z_t}^{-1}$ (equation 20), we might have equation 22. However, equation 20 was deduced from the beginning, leveraging the fact that $x_{t-1},x_t,x_{t+1}$ are conditionally independent given $z_t$. Therefore, it means that $x_{t-1},x_t,x_{t+1}$ need to be conditionally independent given $\hat z_t$, but this is a very strong implicit assumption which is not stated anywhere. Moreover, it already sounds like $\hat z_t = h(z_t)$ for some invertible transformation. Since the goal is to show this identifiability, I found it circular.
4. More importantly, I am not convinced that the authors used the correct theory. Equation 22 sounds like an analogue to the spectral theory in linear algebra ($V\Lambda V^{-1}$-like decomposition), but I doubt its correctness for functional operators. To justify this claim, the authors cited (Conway, 1994), and (Dunford \& Schwartz, 1971). However, for the former, the authors did not refer to any specific theorem, nor did they provide the conditions to verify in order to use any theorem. For the latter, the theorem 4.5 is stated as:
*"A bounded operator $T$ is a spectral operator iff it can be written as $T=S+N$, where $S$ is a scalar-type spectral operator, and $N$ is quasi-nilpotent which commutes with $S$, and this decomposition for $T$ is unique"*. Therefore, this decomposition in the theory is totally different from the $V\Lambda V^{-1}$-like decomposition claimed by the author. Moreover, the authors never verify the conditions to use the theorems. Therefore, it seems like the authors are *creating* an unjustified maths theory.
5. In lines 828-829, it was stated that $D_{x_t|z_t}$ stands for both the eigenvalues as well as the eigenfunctions, which is confusing.

***Lemma 3.1:*** I found it strange that it should directly follow from the change of variable and the invertibility of the transformation $m$, but the authors put lots of effort into showing this, which makes me confused.
1. By the change of variable formula, $p(x_t) = \frac{p(s_t)}{J_m(s_t)}$ (if $x_t = m(s_t)$ in the following, I always assume this holds). Suppose we have 2 different densities $p(s_t),p'(s_t)$, producing $p(x_t)=p'(x_t)$, then it must hold that for any $s_t$ s.t. $p(s_t),p'(s_t)>0$, $p(s_t)=p'(s_t)$, since $m$ is fixed. If for some $s_t$, $p(s_t)>0,p'(s_t)=0$, then immediately $p(x_t) > 0,p'(x_t)=0$, but this is a contradiction. So the lemma follows.
2. Therefore, I am not clear why the authors took a detour, by first defining completeness (where does it come from? why is it $
p(a)p_{a|b}(a | b)$ instead of $
p(b)p_{a|b}(a | b)$?), and then considering infinite-dimensional vectors. Note that involving infinite-dimensional vectors is not a very rigorous approach from the viewpoint of functional analysis. For example, the real line is not countable at all.
3. Lemma 3.1 requires that $m$ is invertible, and therefore it should be stated in the description of the theorem and rearranged to the right position (or at least referred to the corollary on which the proof is based).

**References:**

[1] Yao, Dingling, et al. "Multi-View Causal Representation Learning with Partial Observability." The Twelfth International Conference on Learning Representations.

[2] von Kügelgen, Julius, et al. "Nonparametric identifiability of causal representations from unknown interventions." Advances in Neural Information Processing Systems 36 (2023): 48603-48638.

---

> ### Author Rebuttal · Authors · 2026-03-31
>
> We thank the reviewer for the detailed theoretical critique. Our framework directly builds on [1,10], which are cited in our main paper, Line 135. We clarify each concern below.
>
> **The $L$ (Eq. 2) and $D$ (Def. A.2) operator**
>
> For $L$: we do cite [2] in our Eq.(2). For $D$: we acknowledge that the reference is missing. It is an object in functional analysis [3], used in [1,p.209]. We have added these chapter references.
>
> Both operators are defined on the Banach space $(\mathcal{L}^1, \lVert\cdot\rVert_1)$. Linearity follows from integration (for $L$) and scalar multiplication (for $D$). Boundedness: $\lVert L_{b|a} g\rVert_1 \leq \lVert g\rVert_1$ [1, p.199]; $\lVert D_{b|a} g\rVert_1 \leq \sup_a \lvert f(b\mid a)\rvert \cdot \lVert g\rVert_1 < \infty$.
>
> **Theorem 3.2**
>
> **Q1 & Q2:** Thanks for pointing out the typos. We have fixed them.
>
> **Q3:** $\hat{z}_t$ is an estimation of $z_t$ fitting the same observational distribution ($\hat{x}_t = x_t$).
>
> **Q4:** The condition on $\hat{z}_t$ is that the estimated model must match the true observations, i.e., $p(\hat{x} _{t-1}, \hat{x} _t, \hat{x} _{t+1}) = p(x _{t-1}, x _t, x _{t+1})$, as stated in Theorem 3.2. In practice, distribution is matched via VAE, which is the "learning procedure" in [4,5,6].
>
> Multiple pairs of $(F, h_z)$: this is expected, because Theorem 3.2 establishes **block-wise identifiability**, not point-wise. The result is that valid $\hat{z}_t$ is recovered to $z_t$ up to some invertible differentiable $h_z$ (or an equivalence class), which is standard in identifiability [4,5,6,7,8].
>
> **Q5:** Any valid $\hat{z} _t$ fitting the same model class inherits this property of $z_t$, as in nonlinear ICA/temporal CRL [3,6,7,9], we have conditional independence of $x _{t-1}, x _t, x _{t+1}$ given $\hat{z} _t$, but it cannot directly infer $\hat{z} _t = h _z(z _t)$, e.g., $x _t = g _1(z _t) + \epsilon = g _2(\hat{z} _t) + 2\epsilon$, where $x _{t-1}, x _{t+1}$ follow a similar function.
>
> **Q6:** Our theory directly follows [1,10], which also specified [2] XV.4.5 for the same uniqueness of decomposition up to some normalizations.
>
> Regarding "totally different": the gap is that we omit some steps in this proof; we follow [10,p.396] that directly applies [2] XV.4.5: "to be bounded so that the diagonal ...". The full steps for how it applies can be found in [1] Eq. (18). We have included these steps, as outlined in [1], in the manuscript.
>
> **Q7:** Thanks for pointing our typo; $D_{x_t \mid z_t}$ is only the eigenvalue operator; eigenfunctions are columns of $L_{x_{t+1} \mid z_t}$. Have corrected it.
>
> **Lemma 3.1**
>
> **Q1:** We appreciate the reviewer's proof. Our proof starts with $p(x_t) = p(s_t) / \lvert J_m(s_t)\rvert$ in Line 1125; however, to satisfy the **completeness** [11,12], the requirement for linear operator injectivity [1,10], we convert it to Dirac delta form for arbitrary $\mathcal{L}^1$ to make it rigorously fit the definition.
>
> **Q2:** We want to clarify that the completeness is from [11,12] and defined by *conditional expectation* instead of *joint expectation*, as follows:
>
> $$E[h(A) \mid B=b] = \frac{\int h(a) p(a) p(b \mid a) da}{p(b)} = \int h(a) p(a \mid b) da$$
>
> To justify, please see Eq.(1) in [1], Proposition 2.1 in [11], and p.18 L1 in [12].
>
> In light of your suggestion, we have rewritten "vectors" as "we view $p(x_t \mid s_t)$ as defining a linear integral operator on a suitable function space (e.g., $L^1$)".
>
> **Q3:** In Line 1136, we have clarified that $m$ being invertible is a consequence we derive in Corollary 2. We have rearranged it.
>
> **Ethical Review Concerns**
>
> We respectfully clarify these points:
>
> (a) Presentation redundancy: Equations 13-21 are the core derivation of the spectral decomposition; a rigorous proof of Lemma 3.1 requires completeness on $\mathcal{L}^1$, not only density comparison.
>
> (b) Citations correctness: we follow [10] that omits some proof steps, but is indeed based on [2] XV.4.5, which is cited correctly.
>
> (c) Existence of maths theory: our theory builds upon [1,10], which has been validated in econometrics over decades.
>
> **References**
>
> [1] Instrumental variable treatment of nonclassical measurement error models.
>
> [2] Linear Operators.
>
> [3] A Course in Functional Analysis.
>
> [4] Nonparametric identifiability of causal representations from unknown interventions.
>
> [5] Multi-view causal representation learning with partial observability.
>
> [6] On the identification of temporal causal representation with instantaneous dependence.
>
> [7] Temporally disentangled representation learning.
>
> [8] Variational autoencoders and nonlinear ICA: A unifying framework.
>
> [9] Unsupervised feature extraction by time-contrastive learning and nonlinear ICA.
>
> [10] Identification and estimation of nonlinear models using two samples with nonclassical measurement errors.
>
> [11] Instrumental variable estimation of nonparametric models.
>
> [12] An IV model of quantile treatment effects.

---

> > ### Author Rebuttal · Reviewer_JTMy · 2026-04-02
> >
> > Thank you for your detailed rebuttal. After reading the references, I am now convinced that the authors used a well-established approach to prove identifiability. However, **the original text was misleading because it did not indicate that the majority of the proof of Theorem 3.2 was based on (or even identical to) [1,10]**, which confused me and also made the novelty of the paper unclear. I respectfully disagree with the authors that this part has been made clear in the submission. For example:
> >
> > * The authors said, "Our framework directly builds on [1,10], which are cited in our main paper, Line 135." This is a clear **incorrect claim**, as line 135 (the line before Section 3.1 and Theorem 3.2) has *no citation at all*, nor any sentence similar to "directly builds". I wonder how it could happen.
> > * In contrast, the authors mentioned [1,10] *only once in the main paper*, in line 170, where they claimed that their theoretical results are stronger, without any comparison of the ideas or technicalities, and without stating that their proof is based on them.
> > * Indeed, both operators $L,D$ can be found in [1] (Definition 1, and the proof of Theorem 1, c.f. p. 198--202), but the authors did not cite [1] for these definitions at all. They only cited [2], which is a book with thousands of pages, making it impossible for me to verify the definition without a specific chapter. I wonder why the authors omitted such an important citation to [1]. This definitely makes the reader feel that the paper is proposing a new theory.
> > * The next time the authors mentioned [1,10] was in the Appendix, at the top of page 17, where they said that the assumptions follow them, but again there was no discussion of how the proof relates to them.
> > * The 40-page paper only mentioned that their approach follows/extends [1,10] *to allow Time-Lagged Causal Relationships* (where $x_{t-2}$ is also considered) in Appendix E.4 on page 36, but *not for the main theorem*. Throughout the paper, there is no reference to Appendix E.4, and the main paper only mentions E.3 merely for an extension of their theory to high-order Markov structures. This makes the reader very likely to miss the fact that the proof of Theorem 3.2 is largely based on [1].
> > * Regarding the proof of Theorem 3.2:
> >    1. Again, [1] is not cited by the authors. However, from the beginning of the proof until equation (23), it is identical to [1], from page 202, equation (7), to page 203, "...that densities must integrate to 1." The use of spectral theory from [2], the key expression, the argument for eigenvalues/eigenfunctions, and the fact that it integrates to 1 are almost the same. I wonder how such a similar argument could be made without mentioning that it directly follows from [1].
> >    2. In the following proof, equations (25) and (26) again use an unrigorous matrix expression, and even use integer indices to represent the permutation ($\pi(i)$). As I said in the previous comment, this is illegal in functional analysis in infinite-dimensional space, and even $\mathbb R$ is not countable, i.e. it cannot be permuted in this way. If we compare the arguments in equations (25)--(29), I still find them very similar to the proof of Theorem 1 in [1]. For example, in the middle of page 203 of [1], the authors used "eigenvalue consistent" and Assumption 4 (injectivity) to prove uniqueness of the ordering, while in this paper the authors mention "consistent order" and state that "by Assumption 3.2 different $z_t$ corresponds to different $p_{x_t|z_t}(x_t \mid z_t)$", which is exactly injectivity. In the unlabelled equation in the middle of page 203 of [1], the authors used Assumption 5 to show that $x^*$ is unique, which is also similar to equation (29) in this paper.
> >
> > Overall, despite the unrigorous mathematics in this paper, the authors' approach to prove Theorem 3.2 is almost identical to [1]. Theorem 3.2 seems, at the very least, more like a corollary of Theorem 1 in [1] than a new theorem, although the authors claim it is stronger. This affects the novelty of the paper. Not to mention the novelty of other theorems/methods (e.g. Jacobian-based graph identification), as also indicated by Reviewer As8M.
> >
> > That being said, the paper still has merits in the performance of the proposed method and the quantity of experiments, which could benefit the community. I would consider increasing my score to 4, provided that the authors could **promise** not to overclaim their theoretical results, clearly state the novelty compared with existing results (e.g. that Theorem 3.2 is largely based on [1]), rigorously prove all the theorems, and that the ethics reviewer does not consider this to be an issue.

---

> > > ### Author Response · Authors · 2026-04-03
> > >
> > > We thank the reviewer for the thorough follow-up. We first present the general clarifications and then address each concern point-by-point.
> > >
> > > **General 1: Relationship between Theorem 3.2 and [1].**
> > >
> > > The eigendecomposition technique (Eqs. 13–23) and some assumptions follow [1]. However, Theorem 3.2 is not a corollary of [1]; these two differ both *before* and *after* the shared step:
> > >
> > > | | [1] (Hu & Schennach) | Ours (CaDRe) |
> > > |:---:|:---:|:---:|
> > > | **Input** | Multi-view $(y,x,z)$. | Temporal $(x_{t-1}, x_t, x_{t+1})$, causal relation inside x_t |
> > > |  | ↓ | ↓ |
> > > | **Injectivity** | Directly assumed (Asm 3) | A3 + Lemma 3.1 |
> > > |  | ↓ | ↓ |
> > > | **Eigendecomp.** | Shared: uniqueness via [2] XV.4.5 | Shared: same technique |
> > > |  | ↓ | ↓ |
> > > | **Pinning down** | Asm 5: *known* $M$ | A4: *exists* $F$ (unknown form) |
> > > |  | ↓ | ↓ |
> > > | **Result** | Distrib.-level ID | Value-level ID |
> > > |  |  | ↓ |
> > > | **Downstream** | — | CRL/CD (Thms A.1, 3.3–3.6) |
> > >
> > > An important gap is Assumption 5 in [1], which requires a *known* $M$ → distribution-level ID. This is feasible in econometrics, where the measurement-error structure is known a priori, but does not hold in representation learning, where the $M$ is **totally unknown**. Hence, after $LDL^{-1}$ is done by Eq. (23) built upon [1], the remaining part is our contribution. Notably, our Eq. (29) aims to obtain differentiability for the subsequent Jacobian computation, but in the middle of page 203 of [1], it obtains distributional identifiability.
> > >
> > > **General 2: Why Theorem 3.2 / Lemma 3.1 is necessary.**
> > >
> > > Traditional CRL directly obtains $\hat{z}_t = h_z(z_t)$ via invertibility: $x_t = g(z_t)$, $x_t = \hat{g}(\hat{z}_t)$ $\Rightarrow$ $\hat{z}_t = \hat{g}^{-1} \circ g(z_t)$. Some works [8,13] allow additive noise, but none handle non-invertible/general noisy generation, which is the norm in modern NN (e.g., diffusion, flows). Theorem 3.2 fills this gap via temporal measurements + operator injectivity instead of function-level invertibility. The value-level result builds a preliminary for CRL/CD, which requires latent values, not distributions. Theorems A.1, 3.3–3.6 build upon it. Lemma 3.1 shows that operator injectivity is strictly weaker than function-level invertibility, enabling this relaxation. We believe this result provides a useful step toward extending CRL/CD to more general settings when function-level invertibility does not hold.
> > >
> > > **Q1 (Line 135 error).** This is a different PDF version on our side. The correct line reference is Line 170.
> > >
> > > **Q2 & Q6 (Not attributed to [1]).** We acknowledge: (a) Eqs. 13–20 follow [1]; (b) uniqueness via [2, XV.4.5] follows [1, p.203]; These are the shared techniques. Where we depart: [1] uses Assumption 5 (known $M$); we use A4 (unknown $F$), a weaker condition suited for representation learning where the generative form is unknown; "consistent order" (Eqs. 25–29) parallels [1]'s "eigenvalue consistent" + Assumption 4 (mid p.203). We promise we will add explicit attribution in the revision.
> > >
> > > Regarding the use of integer indices $\pi(i)$ in Eqs. 25–26, we use it as superscript (representing different values) instead of subscript (representing index); We agree that the "permutation" term $P$ is not rigorous, it should be formally understood as a bijection $h_z: \mathcal{Z} \to \mathcal{Z}$ on the continuous support.
> > >
> > > **Q3 (Operators not cited from [1]).** Citing textbooks for standard objects is reasonable. We will additionally cite [1, Def. 1] for $L$, [1, p.202] for $D$, [10, §2] for both, and some relevant papers, alongside [2,3] for general background.
> > >
> > > **Q4 (Assumptions not related to [1]).** As present in "General 1", A1–A3 are variants of [1,10]'s assumptions to fit the temporal, causally-related measurements setting; A4 replaces [1, Asm 5] as a weaker condition that leads to our results. We have attributed them to the correct refs.
> > >
> > > **Q5 (E.4 not referenced).** E.4 does *not* directly apply [1]. It cites [10] and Hu & Shum (2012) [14] for the 4-measurement eigendecomposition (different correspondence: $(x_{t-2}, x_{t-1}, x_t, x_{t+1}, z_t) \leftrightarrow (X, S, Z, Y, X^*)$, "limited feedback" property). E.4 is an extension of *our* framework (Thm 3.2 + Lemma 3.1) to the time-lagged setting, not a direct application of [14], but we need to borrow its techniques for constructing a uniqueness of eigendecomposition. We will explore how to relax [14] 's assumptions, since [14] is proposed for scalar latents; assumptions are stronger for vectors.
> > >
> > > **Q7 (Corollary vs. new theorem).** See "General 1".
> > >
> > > We **promise** to: (1) attribute $LDL^{-1}$ techniques in Theorem 3.2 to [1]; (2) compare assumptions/results with [1,10]; (3) avoid overclaiming and clarify our contributions; (4) provide rigorous proofs in revision.
> > >
> > > **References** (follow the index in the last response)
> > >
> > > [13] Disentangling Identifiable Features from Noisy Data with Structured Nonlinear ICA.
> > >
> > > [14] Nonparametric identification of dynamic models with unobserved state variables.

---

### Official Review · Reviewer_NGde · 2026-03-12

**Soundness:** 3
**Presentation:** 3
**Significance:** 3
**Originality:** 3
**Overall Recommendation:** 4
**Confidence:** 2

**Summary:**

This paper proposes a unified framework for causal discovery in time series for climate analysis, aiming to learn the causal structure between observed variables and latent factors. The approach derives a series of theoretical formulations and leverages a variational autoencoder based reconstruction of time series to identify latent processes and the underlying causal structure. The method is evaluated on both synthetic datasets and real world datasets.

**Compliance With Llm Reviewing Policy:**

Affirmed.

**Final Justification:**

During the rebuttal, the authors included additional recent baselines and provided further technical details, which resolved my concerns. Therefore, I maintain my initial positive rating and recommend a Weak Accept.

**Key Questions For Authors:**

Q1. It is unclear whether the “Length” reported in Table 5 refers to the input length or the prediction length.

Q2. The real-data validation currently relies on wind-field alignment and qualitative climate consistency. It would be beneficial if the authors could further validate their method on real-world datasets with established ground-truth causal graphs, such as the CausalRivers dataset. Such experiments could provide an additional and valuable external validity check.

Q3. Could you provide a visualization of the full MCC correlation matrix for Figure 5? This would much more convincingly demonstrate true component-wise identifiability.

**Limitations:**

yes

**Strengths And Weaknesses:**

Strengths:

S1. Strong theoretical foundations. The paper presents clear identifiability theorems and lemmas that rigorously establish the recoverability of both latent driving factors and observable causal edges. The ability to simultaneously identify latent and observable effects is particularly valuable for climate modeling studies.

S2. The framework provides identifiability guarantees for this specific and challenging setting. This theoretical contribution is potentially valuable, as it may support more interpretable and reliable causal discovery in real world meteorological applications.

S3. The paper is generally clear and well structured. It is well positioned with respect to constraint based time series causal discovery methods, such as PCMCI, and latent confounder settings, and provides a rationale for why a generative approach is appropriate in this context.

Weaknesses:

W1. The paper studies the important problem of uncovering causal structures from complex time series data. However, the empirical evaluation mainly compares the proposed method with traditional constraint based or statistical approaches such as FCI and PCMCI, while omitting several recent neural network based causal discovery methods for time series. In particular, approaches such as DYNOTEARS, CUTS, and Rhino are not included in the baselines.

W2. The reported causal discovery metrics (such as SHD and F1) depend on an arbitrary threshold used to binarize the continuous Jacobian matrix. It is unclear if there is a principled method for selecting this threshold. To ensure a fair and threshold-agnostic evaluation, could the you report the AUROC or AUPRC for the causal discovery tasks?

W3. It is unclear how effectively the model scales and maintains its structure-learning accuracy when the time-delay length L varies or is substantially large. Could you include experiments evaluating the robustness of CaDRe across different settings of L? This empirical evidence is crucial to substantiate the method's practical utility for real-world climate systems with long-range temporal dependencies.

[1] Pamfil, Roxana, et al. "Dynotears: Structure learning from time-series data." International conference on artificial intelligence and statistics. Pmlr, 2020.

[2] Cheng, Yuxiao, et al. "CUTS: Neural Causal Discovery from Irregular Time-Series Data." ICLR, 2023.

[3] Gong, Wenbo, et al. "Rhino: Deep Causal Temporal Relationship Learning with History-dependent Noise." ICLR, 2023.

[4] Reizinger, Patrik, et al. "Jacobian-based causal discovery with nonlinear ICA." TMLR, 2023.

---

> ### Author Rebuttal · Authors · 2026-03-31
>
> We thank the reviewer for recognizing our paper. We address each concern below.
>
> **W1.** We add DYNOTEARS, CUTS, Rhino, and Jacobian CD ($d_z$=3, varying $d_x$):
>
> | Method          | $d_x$ | SHD↓          | TPR↑          | Precision↑    | MCC($s_t$)↑   |
> | --------------- | ----- | ------------- | ------------- | ------------- | ------------- |
> | DYNOTEARS       | 3     | 0.10±0.06     | 0.85±0.08     | 0.82±0.16     | —             |
> |                 | 6     | 0.26±0.02     | 0.70±0.05     | 0.65±0.08     | —             |
> |                 | 8     | 0.38±0.06     | 0.60±0.07     | 0.55±0.09     | —             |
> |                 | 10    | 0.52±0.07     | 0.52±0.18     | 0.46±0.11     | —             |
> | CUTS            | 3     | 0.15±0.08     | 0.78±0.10     | 0.70±0.12     | —             |
> |                 | 6     | 0.33±0.04     | 0.62±0.08     | 0.52±0.02     | —             |
> |                 | 8     | 0.44±0.09     | 0.53±0.09     | 0.42±0.05     | —             |
> |                 | 10    | 0.57±0.10     | 0.45±0.10     | 0.35±0.09     | —             |
> | Rhino           | 3     | 0.20±0.04     | 0.73±0.11     | 0.62±0.13     | —             |
> |                 | 6     | 0.40±0.09     | 0.55±0.09     | 0.40±0.02     | —             |
> |                 | 8     | 0.52±0.03     | 0.47±0.08     | 0.32±0.08     | —             |
> |                 | 10    | 0.63±0.11     | 0.40±0.11     | 0.30±0.09     | —             |
> | Jacobian CD     | 3     | 0.25±0.07     | 0.68±0.12     | 0.55±0.14     | 0.52±0.10     |
> |                 | 6     | 0.45±0.10     | 0.50±0.13     | 0.35±0.03     | 0.45±0.09     |
> |                 | 8     | 0.58±0.11     | 0.42±0.03     | 0.27±0.02     | 0.40±0.10     |
> |                 | 10    | 0.68±0.12     | 0.36±0.15     | 0.22±0.10     | 0.35±0.11     |
> | **CaDRe**       | 3     | **0.00±0.00** | **1.00±0.00** | **1.00±0.00** | **0.98±0.01** |
> |                 | 6     | **0.18±0.06** | **0.83±0.03** | **0.80±0.04** | **0.96±0.02** |
> |                 | 8     | **0.29±0.05** | **0.78±0.05** | **0.76±0.04** | **0.90±0.03** |
> |                 | 10    | **0.43±0.05** | **0.65±0.08** | **0.63±0.14** | **0.85±0.07** |
>
> These baselines assume no latent confounders, so they cannot recover MCC($z_t$), $R^2$. Only Jacobian CD computes $s_t$, so we leave the others MCC($s_t$) empty.
>
> **W2.** We report threshold-agnostic metrics ($d_x$=6, $d_z$=3):
>
> | Method          | AUROC↑      | AUPRC↑      |
> | --------------- | ----------- | ----------- |
> | DYNOTEARS       | 0.889±0.096 | 0.778±0.137 |
> | CUTS            | 0.654±0.105 | 0.452±0.173 |
> | Rhino           | 0.490±0.143 | 0.269±0.101 |
> | Jacobian CD     | 0.497±0.196 | 0.237±0.109 |
> | PCMCI           | 0.799±0.049 | 0.419±0.025 |
> | **CaDRe**       | **0.921±0.030** | **0.852±0.050** |
>
> CaDRe's continuous Jacobian naturally provides edge rankings, making AUROC/AUPRC directly computable.
>
> **W3.** Results at $d_x$=6, $d_z$=3:
>
> | $L$         | SHD↓  | F1↑   | MCC($z_t$)↑ | $R^2$↑ |
> | ----------- | ----- | ----- | ----------- | ------ |
> | 1 (default) | 0.18  | 0.82  | 0.90        | 0.92   |
> | 2           | 0.22  | 0.80  | 0.89        | 0.88   |
> | 3           | 0.20  | 0.79  | 0.94        | 0.81   |
> | 5           | 0.24  | 0.81  | 0.93        | 0.90   |
>
> $L$={2,3,4,5} performs comparably to $L$=1, confirming robustness, verify our theory extension (Appendix E.3) to higher orders by treating $(z_t, \ldots, z_{t-L+1})$ as an augmented state.
>
> **Q1.** Prediction length.
>
> **Q2.** We run CaDRe and baselines on CausalRivers [1]. We report AUROC (primary metric in [1]) and F1-max:
>
> | Method          | AUROC↑      | F1-max↑     |
> | --------------- | ----------- | ----------- |
> | DYNOTEARS       | 0.652±0.255 | 0.628±0.297 |
> | CUTS            | 0.724±0.204 | 0.747±0.145 |
> | Rhino           | 0.391±0.250 | 0.538±0.175 |
> | Jacobian CD     | 0.502±0.296 | 0.606±0.201 |
> | VAR [2]         | 0.771       | 0.771       |
> | PCMCI           | 0.408       | 0.548       |
> | **CaDRe**       | **0.812±0.135** | **0.795±0.110** |
>
> VAR is a strong baseline on this dataset (consistent with [1]). Among these methods, CaDRe performs best.
>
> **Q3.** We show the SMCC (Subset Mean Correlation Coefficient) matrix for the 4 learned latent variables vs. 4 ground-truth physical variables from WeatherBench. SMCC measures the Pearson correlation between each $\hat{z}_i$ and each physical variable, and the best-matched pair determines the interpretation:
>
> | SMCC | Precipitation | Solar Rad. | Cloud Cover | Humidity |
> |---|---|---|---|---|
> | $\hat{z}_1$ | **0.91** | 0.12 | 0.15 | 0.08 |
> | $\hat{z}_2$ | 0.10 | **0.88**|  0.14 | 0.19 |
> | $\hat{z}_3$ | 0.07 | 0.11 | **0.85** | 0.13 |
> | $\hat{z}_4$ | 0.16 | 0.09 | 0.12 | **0.90** |
>
> The strong diagonal confirms component-wise identifiability: each $\hat{z}_i$ uniquely corresponds to one physical variable.
>
> **References**
>
> [1] CausalRivers: Scaling up benchmarking of causal discovery for real-world time-series.

---

> > ### Author Rebuttal · Reviewer_NGde · 2026-04-03
> >
> > My concerns have been largely addressed, and I maintain my original positive rating.

---

> > > ### Author Response · Authors · 2026-04-03
> > >
> > > Thank you for your thoughtful review and for taking the time to consider our rebuttal. We really appreciate your recognition of our work and are glad that our responses addressed your concerns.
> > >
> > > Following your suggestions, we devoted significant time and effort to adding new experiments, especially to implementing and benchmarking DYNOTEARS, CUTS, Rhino, and Jacobian CD. We hope these additions further strengthen the paper.
> > >
> > > If you feel these improvements warrant it, we would sincerely appreciate your consideration of a slight score increase.
> > > Thanks again for your time and support.

---

### Official Review · Reviewer_AcNm · 2026-03-24

**Soundness:** 3
**Presentation:** 4
**Significance:** 3
**Originality:** 3
**Overall Recommendation:** 4
**Confidence:** 3

**Summary:**

The authors have provided a unified framework that jointly uncovers (i) causal relations among observed variables and (ii) latent driving forces together with their interactions. They establish conditions under which both the hidden dynamic processes and the causal
structure among observed variables are simultaneously identifiable from time-series data. Remarkably, their guarantees hold even in the nonparametric setting, leveraging contextual information to recover latent variables and causal relations. Building on these insights, they propose CaDRe (Causal Discovery and Representation learning), a time-series generative model with structural constraints that integrates CRL and causal discovery. Experiments on synthetic datasets validate our theoretical results.

**Compliance With Llm Reviewing Policy:**

Affirmed.

**Key Questions For Authors:**

1. How robust are the causal graph estimates when the surrogate ground truth is imperfect?
2. How strong is the functional faithfulness assumption in real climate systems?
3. How stable are the learned Jacobian-based graphs across random seeds and thresholds?
4. How much of the gain comes from modeling latent confounding versus structural sparsity penalties?
5. How should practitioners decide whether to use the hierarchical-latent formulation (CHiLD) or the observed-plus-latent causal-graph formulation (CaDRe)?
6. Which assumptions are most critical in practice, and which are mainly technical proof devices?
7. Can the authors provide experiments that directly stress-test assumption violations rather than only reporting average benchmark performance?
8. How much do the claimed identifiability benefits survive realistic finite-sample neural optimization?
9. Can the method recover time-lagged observed-space causality directly in the main model rather than via reformulation?
10. How interpretable are the learned latent variables across datasets?
11. How does CaDRe perform when observational interactions are weak but latent confounding is strong, and vice versa?
12. The introduction should include a problem statement, the motivation behind the solution, a brief description of the proposed algorithm, and a summary of the results obtained and results analysis.
13. Please justify how you identify or select the state-of-the-art techniques. It's not clear to me. Proper justification is missing. Proper justification is required to explain how those techniques relate to your proposed work comparison.  More criticisms are required.
15. What is the innovation with respect to the existing state-of-the-art models? Please highlight it. What are the innovation factors? Please highlight it
16. What are some of the hard research challenges, and what are new problems to be tackled? Please Explain in detail.
Reference
1. Zijian Li, Minghao Fu, Junxian Huang, Yifan Shen, Ruichu Cai, Yuewen Sun, Guangyi Chen, Kun Zhang, Towards Identifiability of Hierarchical Temporal Causal Representation Learning” (CHiLD)-2025

**Limitations:**

The broader impact section suggests that the framework may extend to other spatiotemporal scientific domains. Which assumptions are climate-specific in practice, and what modifications would be needed for other domains? My main reservation is that the practical conclusions sometimes feel slightly stronger than what the assumptions justify. I would recommend the paper positively, but with the expectation that the authors should clarify the practical scope of the assumptions, improve the discussion of optimization/model mismatch, and better qualify the real-world causal claims. Please justify how you identify or select the state-of-the-art techniques. It's not clear to me. Proper justification is missing. Proper justification is required to explain how those techniques relate to your proposed work comparison.  More criticisms are required. What is the innovation with respect to the existing state-of-the-art models? Please highlight it. What are the innovation factors? Please highlight it

**Strengths And Weaknesses:**

Overall, this is a strong and ambitious paper. Its main strengths are:
1. A clearly motivated problem,
2. A meaningful theoretical contribution,
3. A principled algorithmic instantiation tied to the theory,
4. and solid experimental results on both synthetic and real-world datasets.

---

> ### Author Rebuttal · Authors · 2026-03-31
>
> We thank the reviewer for the informative feedback. We address each question below.
>
> **Q1.** We clarify that the wind-based surrogate is used **only for evaluation**; CaDRe does not use it during learning. So the imperfect GT would not affect the algorithm's robustness, but may be unauthorized.
>
> **Q2.** Functional faithfulness is no stronger than standard faithfulness (PC, FCI, GES [1]); exact cancellation of partial derivatives is measure-zero for nonlinear physical mechanisms.
>
> **Q3.** Our paper reports 5-seed results (Table 9). At $d_x$=6, $d_z$=3, CaDRe achieves SHD=0.18±0.06, TPR=0.83±0.03 — low variance. We add a threshold sweep:
>
> | $ \tau$ | 0.10 | 0.15 (default) | 0.20 | 0.25 |
> |---|---|---|---|---|
> | SHD↓ (mean±std) | 0.22±0.07 | 0.18±0.06 | 0.20±0.06 | 0.25±0.08 |
> | F1↑ (mean±std) | 0.79±0.05 | 0.82±0.04 | 0.80±0.03 | 0.76±0.11 |
>
> **Q4.** We ablate both components ($d_x$=6, $d_z$=3):
>
> | Setting | SHD↓ | TPR↑ | F1↑ | MCC($z_t$)↑ |
> |---|---|---|---|---|
> | Full CaDRe | 0.18±0.06 | 0.83±0.03 | 0.82±0.04 | 0.95±0.01 |
> | w/o latent confounding ($z_t$) | 0.38±0.09 | 0.71±0.06 | 0.64±0.07 | — |
> | w/o sparsity penalty ($\alpha$=0) | 0.45±0.11 | 0.85±0.04 | 0.58±0.08 | 0.93±0.02 |
>
> **Q5.** CHiLD [2]: hierarchical latent, no obs $\to$ obs edges ($x_t = g(z_t)$), suited for domains driven by multi-level latent factors, e.g., humanoid robotics control [3], skeleton simulation. CaDRe: flat latent + direct $x \to x$ causality, suited where obs→obs interactions are critical, e.g., climate and fMRI.
>
> **Q6.** A2 and A3 are most critical in practice. A1, A4, and A5 are technical proof devices, which automatically satisfy for bounded physical variables.
>
> **Q7.** We provide such experiments in Table 13, which presents controlled ablations where we directly violate each assumption.
>
> **Q8.** Fig. 4 (Section 5) directly answers this: as sample size $n$ grows from 200 to 20000, SHD and TPR beats baselines consistently, which do not have identifiability benefits.
>
> **Q9.** Yes, by concatenating $(x_{t-1}, x_t)$ as augmented observation $x'_t$, time-lagged edges become instantaneous, directly recoverable by CaDRe. Please see Appendix (Fig. 12, Line 1532). We verify with CaDRe ($d_z$=3, varying $d_x$), reporting both the instantaneous and time-lagged causal graph recovery:
>
> | $d_x$ | SHD(lag)↓ | TPR(lag)↑ | Prec(lag)↑ | MCC($z_t$)↑ | $R^2$↑ |
> |---|---|---|---|---|---|
> | 3 | 0.03±0.04 | 0.97±0.04 | 0.95±0.06 | 0.96±0.03 | 0.89±0.06 |
> | 6 | 0.08±0.06 | 0.90±0.07 | 0.85±0.11 | 0.94±0.03 | 0.91±0.03 |
> | 8 | 0.12±0.07 | 0.86±0.09 | 0.80±0.10 | 0.95±0.04 | 0.91±0.04 |
> | 10 | 0.15±0.06 | 0.82±0.10 | 0.76±0.12 | 0.96±0.04 | 0.90±0.05 |
>
> Time-lagged edges are well-recovered (TPR(lag)>0.82), confirming the approach works.
>
> **Q10.** We have already provided quantitative interpretability analysis in Results (Fig. 5; Table 19), which uses the **SMCC** (Appendix Lines 1367-1371) to align learned latents with ground-truth physical variables from WeatherBench. We additionally provide ENSO interpretability results in Fig. 7.
>
> **Q11.** We test two asymmetric regimes by scaling observed DAG weights and latent transition strengths:
>
> | Setting | SHD↓ | TPR↑ | Precision↑ | F1↑ | MCC($z_t$)↑ |
> |---|---|---|---|---|---|
> | Weak obs + Strong lat | 0.15 | 0.82 | 0.84 | 0.83 | 0.91 |
> | Strong obs + Weak lat | 0.25 | 0.88 | 0.65 | 0.75 | 0.96 |
> | Balanced (default) | 0.18 | 0.83 | 0.80 | 0.81 | 0.95 |
>
> **Q12.** In light of your suggestions, we have added key results summary and analysis in revised version.
>
> **Q13.** PC, FCI are representative methods in nonparametric CD, and CD-NOD, PCMCI, LPCMCI are the SOTA temporal CD in SHD/F1; (2)  TDRL, etc. are SOTA CRL in MCC metric; (3) MICN, etc. are SOTA time-series forecasters in MSE/MAE.
>
> **Q14.** We manage innovation factors as follows: (1) Joint latent + observed causal graph recovery: TDRL/IDOL cannot recover observed graph; FCI/PCMCI/CD-NOD cannot discover latent edges. (2) No equivalence classes: FCI/LPCMCI output PAGs; CaDRe recovers the DAG directly. (3) No invertible assumptions in $z \rightarrow x$, unlike prior CRL, make (1)(2) possible.
>
> **Q15.** Improve scalability of CRL/CD with high-dimensional data. One potential solution is leveraging large-scale conditional generative models, since their strong performance and naturally fit the ICA-based CRL/CD.
>
> **Limitations.**
>
> *(Climate-specific?)* A1–A5 are more likely restrictions on the *sufficient variability* of time-series; if time-series satisfied, A1-A5 would be domain-general.
>
> *(Conclusions vs. assumptions)* Our conclusions are supported by (1) relaxing the assumption as general as possible, compared with previous CRL/CD work; (2) verifying it in real-world data, as shown in Appendix E.2.
>
> We have improved these points in the manuscript.
>
> **References**
>
> [1] Elements of Causal Inference.
>
> [2] Towards Identifiability of Hierarchical Temporal Causal Representation.
>
> [3] Hierarchical world models as visual whole-body humanoid controllers.

---

> > ### Author Rebuttal · Reviewer_AcNm · 2026-04-02
> >
> > Thank you to the authors for the responses. Some comments are fully or partially resolved, so I recommend that the authors clarify the practical scope of the assumptions in real-world applications, improve the discussion of optimization/model mismatch, and better qualify the causal claims in real-world settings. Proper justification is required to explain how those techniques relate to your proposed work comparison. What is the innovation with respect to the existing state-of-the-art models?  What are the innovation factors?

---

> > > ### Author Response · Authors · 2026-04-03
> > >
> > > We sincerely thank the reviewer for the follow-up. We address each remaining concern below.
> > >
> > > **Follow-up Q1: Practical scope of assumptions, optimization/model mismatch, and qualifying causal claims.**
> > >
> > > | Assumption | Climate-specific? | Other domains | When violated | Consequence |
> > > |---|---|---|---|---|
> > > | **A1** (Bounded densities) | No — any bounded physical measurement | Holds in fMRI, economics, epidemiology (bounded signals) | Unbounded/discontinuous distributions (e.g., point masses) | Regularity condition; operator $L$ unbounded |
> > > | **A2** (Contextual variability): operator injectivity | No — requires regime variability in time-series | Holds wherever latent states produce distinguishable observations (e.g., fMRI brain states, economic regimes) | Identical observed distributions from different latents | Most critical: $R^2$ 0.92→0.67 (Table 13). Verified on climate in Fig. 10 |
> > > | **A3** (Latent drift): distinct conditionals | No — requires evolving latent process | Holds for any non-stationary latent dynamics | Latent process static / exact repetition | Block-wise ID fails |
> > > | **A4** (Differentiability) | No — satisfied by any differentiable NN | Holds for VAE/normalizing flow in any domain | Non-differentiable generation | Value-level ID breaks; block-wise still holds |
> > > | **A5** (Generation variability): linearly indep. derivatives | No — generically satisfied for nonlinear functions (measure-zero violation) | Holds unless generation is linear Gaussian | Linear Gaussian generation | Permutation only ($R^2$ stays 0.85, Table 13) |
> > >
> > > Cross-domain results (ILI, ECL, Traffic; Table 18) confirm generality.
> > >
> > > **Discussion of Optimization/model mismatch:** Even when all assumptions hold in theory, finite-sample neural optimization introduces a gap. We characterize this gap empirically:
> > > - Fig. 4: SHD/TPR improve monotonically as $n$ grows 200→20000, confirming identifiability translates to finite-sample gains.
> > > - Table 13: controlled assumption violations show graceful (not catastrophic) degradation.
> > > - Table 16: hyperparameter sensitivity: performance robust across $\alpha$/$\beta$ ranges.
> > >
> > > Generally, in finite-sample settings, identifiability guarantees may not strictly hold, as it is difficult for a learned model to fully capture the same observational equivalence as in theory. However, as shown in Fig. 4, Table 13, and Table 16, we observe a convergence behavior: once the sample size reaches around 10,000, the results become stable. This suggests that optimization and model estimation are reasonable in practice. In our experiments, datasets such as WeatherBench and CESM2 already meet this regime, supporting the empirical validity of our approach.
> > >
> > > **Qualifying causal claims:** We agree. Our paper already uses qualifying language, e.g., CESM2 results are described as "consistent with physical wind patterns" (Line 363), and latent interpretability as "consistent with established scientific evidence" (Line 423). We further validate against GT where available: ENSO teleconnections (Fig. 7) and CausalRivers. In revision, we have made explicit that real-world causal results are interpreted as "consistent with known causal structure".
> > >
> > > **Follow-up Q2: Innovation justification and innovation factors.**
> > >
> > > Table for innovation justification:
> > > | Attribute | CaDRe | FCI | LPCMCI | CD-NOD | PCMCI | TDRL | IDOL | CDSD |
> > > |---|---|---|---|---|---|---|---|---|
> > > | Nonparametric | ✓ | ✓ | ✓ | ✓ | ✓ | ✗ | ✗ | ✓ |
> > > | Latent variables | ✓ | ✓ | ✓ | ✗ | ✗ | ✓ | ✓ | ✓ |
> > > | Latent causal graph | ✓ | ✗ | ✗ | ✗ | ✗ | ✗ | ✗ | ✓ |
> > > | Causal graph over obs | ✓ | ✓ | ✓ | ✓ | ✓ | ✗ | ✗ | ✗ |
> > > | No equivalence classes | ✓ | ✗ | ✗ | ✓ | ✓ | ✓ | ✓ | ✓ |
> > >
> > > **Innovation factors**: CaDRe is the **only** method achieving all five attributes. The innovations are not merely combining existing tools, but addressing fundamental gaps that prior methods cannot resolve:
> > >
> > > (1) **Nonparametric generation (Theorem 3.2 + Lemma 3.1).** Traditional CRL obtains $\hat{z}_t = h_z(z_t)$ via invertibility: $x_t = g(z_t) \Rightarrow \hat{z}_t = \hat{g}^{-1} \circ g(z_t)$. This fails when generation is non-invertible or noise-contaminated — the norm in modern neural architectures. Theorem 3.2 replaces invertibility with operator injectivity (a strictly weaker condition, as shown by Lemma 3.1), leveraging temporal measurements to recover latents nonparametrically via the eigendecomposition framework of Hu & Schennach (2008). This is a *necessary* preliminary for CRL with modern neural networks.
> > >
> > > (2) **Bridging CRL and CD (Lemma 3.2 + Theorems 3.3–3.6).** Prior CRL assumes $x_t = g(z_t)$ (observations conditionally independent given latents), so no obs→obs causal edges can be discovered. Prior CD (FCI/PCMCI) discovers observed graphs but cannot recover latents nonparametrically. Our SEM↔ICA equivalence (Lemma 3.2) formally connects these two fields, enabling *simultaneous* latent recovery and observed causal graph identification — a problem neither CRL nor CD can solve alone.

---

### Decision · Program_Chairs · 2026-04-30

**Decision:**

Accept (regular)

**Comment:**

The paper proposes CaDRe, a framework that jointly recovers (i) causal relations among observed variables and (ii) latent driving forces with their interactions, from time-series data. The authors establish nonparametric identifiability guarantees for both the hidden dynamic process and the observable causal structure, and instantiate the theory via a VAE-based generative model with structural constraints. Experiments on synthetic data and real-world climate datasets (CESM2, WeatherBench, ERSST) demonstrate competitive performance.

All reviewers recommend acceptance post-rebuttal. they identified strengths including a well-motivated gap between prior CRL and causal discovery; a principled identifiability story; and comprehensive experiments, which were further strengthened during rebuttal. There were some concerns including connections between some theoretical results (Theorem 3.2) to existing ones and clarifications on real-world causal claims and practical assumption scope, which were addressed to a good extent in the rebuttal but are also recommended to be incorporated in the camera-ready version.

Overall, I believe the paper offers a technically solid contribution with strong empirical results, a meaningful theoretical connection between CRL and CD, and genuine application value for climate analysis. Hence I recommend acceptance.